# MATRIX DATA DEEP DECODER - GEOMETRIC LEARNING FOR STRUCTURED DATA COMPLETION

## ABSTRACT

In this work we present a fully convolutional end to end method to reconstruct corrupted sparse matrices of Non-Euclidean data.The classic example for such matrices is recommender systems matrices where the rows/columns represent items/users and the entries are ratings. The method we present is inspired by the surprising and spectacular success of methods like" deep image prior" and "deep decoder" for corrupted image completion. In sharp contrast to previous Matrix Completion methods wherein the latent matrix or its factors directly serve as the optimization variable, in the method we present, the matrix is parametrized as the weights of a graph neural network acting on a random noisy input. Then we are tuning the network parameters to get a result as close as possible to the initial sparse matrix (using it's factors) getting that way state of the art matrix completion result. In addition to the conceptual simplicity of our method, which is just non-Euclidean generalization of deep image priors, it holds less parameters then previously presented methods which makes the parameters more trackable and the method more computationally efficient and more applicable for the real world tasks.The method also achieves state-of-the-art results for the matrix completion task on the classical benchmarks in the field. The method also surprisingly shows that untrained convolutional neural network can use a good prior not only for image completion but also for Matrix Completion when redefined for graphs.

## 1 INTRODUCTION

**Matrix completion** (MC) consists of estimating the missing entries of an $n \times m$ matrix $\boldsymbol{X}$ (usually, of very big dimensions) given its measurements $\boldsymbol{M}$ on a (usually, very sparse) support $\Omega$. An example of such matrices are signals on graphs/manifolds which are Non-Euclidean domains. The classical example of such data are recommender (recommendation) systems, where the ratings are signals on (user item) couple. The most known Matrix Completion problem is the Netflix problem, where a 1M $ prize was offered for the algorithm that can best predict user ratings in a dataset that contained 480k movies $\times$ 18k users (8.5B entries), with $0.011\%$ known entries (Bell et al., 2009).

Many works focused on solutions for the MC problem. In brief, one wishes to obtain the matrix $\boldsymbol{X}$ given matrix $\boldsymbol{M}$ as the specified input on the support $\Omega$. Then, formally the completion task amounts to the minimization problem

$$\widehat{\boldsymbol{X}} = \underset{\boldsymbol{X}}{\operatorname{argmin}} \|\boldsymbol{A}_\Omega \circ (\boldsymbol{X} - \boldsymbol{M})\|_F^2$$

where $\boldsymbol{A}_\Omega$ is the observation mask matrix (filled with 1 where data exists in the original problem), $\circ$ is the Hadamard product and $\|.\|_F^2$ is the Frobenius norms (Rennie & Srebro, 2005). Different approaches where presented in order to fill in matrix $\boldsymbol{X}$. Those approached included imposing different regularization (priors) on the matrix and its factors. The most prominent approach consists of imposing a low rank (Candès & Recht, 2009; Recht, 2009) on the matrix. Then, priors based on *collaborative filtering* (users/items rating patterns), *content based filtering* (user/items profile) (Ghassemi et al., 2018; Jain & Dhillon, 2013; Xu et al., 2013; Si et al., 2016) and their combinations. Then Geometric Matrix Completion approaches appeared (Li & Yeung, 2009; Rao et al., 2015; Cai et al., 2011) and proposed describing rows/column graphs which represent similarity, then encoding the structural (geometric) information of those graphs via graph Laplacian regularization (Belkin & Partha, 2002; Belkin & Niyogi, 2003) and imposing smoothness of the data in those graphs

(Kalofolias et al., 2014; Rao et al., 2015; Ma et al., 2011; Mardani et al., 2012). Those approaches where generally related to the field of signal processing as entries signals on the rows/columns graphs (Shuman et al., 2012). Then Geometric Deep Learning Methods where introduced to learn the domains of geometric data structures (e.g. single graphs or manifolds)(Bronstein et al., 2016; Lefkimmiatis, 2016; Defferrard et al., 2016; Niepert et al., 2016; Gilmer et al., 2017; Hamilton et al., 2017; Velickovic et al., 2017; Chen et al., 2018; W. Huang et al., 2018; Klicpera et al., 2018; Abu-El-Haija et al., 2019; Ying et al., 2018; Gao et al., 2018; Hammond et al., 2011). The current state of the art solution for Matrix completion problem, relies on an extending classical harmonic analysis methods to non-Euclidean domains. When, the geometry of the column/row spaces and their graphs is utilised to provide a Geometric Deep Learning mechanism called the RMGCNN (Monti et al., 2017) that includes a complex combined CNN and RNN(Hochreiter & Schmidhuber, 1997) networks.

In this work we present a simplified method for the MC problem: the Matrix Data Deep Decoder that contains a classical end to end GRAPH convolutional neural network and inspired by the leading methods from the field of image completion - the Deep Image Prior (Ulyanov et al., 2020) and the Deep Decoder (Heckel & Hand, 2018). In our method, random noisy input matrix is acted upon by the weights of a neural network (parametrization). By tuning the parameters of the network and minimising the error between its output to the initial corrupted matrix, we find the best candidate for the complete matrix. This method yields state of art results for the MC task. The contributions of our work are:

- A novel approach for solving the MC Problem, using deep learning with end-to-end pure convolutional network for graphs.

- State-of-the-art performance for the MC problem in both prediction error (RMSE) and solution running time[1]. Our method significantly outperforms the previous state of art method - the RMGCNN.

- We show that a pure graph convolutional neural network is a good prior for the MC problem. This provides a correspondence of convolutional neural networks methods to MC problems.

## 2 PRELIMINARIES

### 2.1 MATRIX COMPLETION NOTATION

The most prominent prior for the MC problem is assuming the matrix $X$ is of **low rank**. Low rank is obtained by rank regularization using its nuclear (trace) norm $\|X\|_*$ – sum of the singular values of $X$. The canonical optimization problem with parameter $\lambda_*$, is stated as:

$$\widehat{X} = \min_{X} \quad \|A_\Omega \circ (X - M)\|_F^2 + \lambda_* \|X\|_*$$

#### 2.1.1 MATRIX FACTORIZATION

To alleviate the computational burden for big datasets, we factorize $X = W H^T$, where $W \in \mathbb{R}^{m \times k}$, $H^T \in \mathbb{R}^{k \times n}$. Here, $k \ll m$ and $n$ is the upper bound on the rank of $X$. With this factorization, the nuclear norm term can be replaced by the sum of the Frobenius norms leading to the following non-convex (but still very well-behaved) problem (Rao et al., 2015):

$$\widehat{X} = \min_{W, H^T} \quad \left\|A_\Omega \circ \left(W H^T - M\right)\right\|_F^2 + \frac{\lambda_*}{2} \left(\|W\|_F^2 + \left\|H^T\right\|_F^2\right)$$

### 2.2 GEOMETRIC MATRIX COMPLETION

We introduce the geometric matrix completion framework, using notations as in RMGCNN (Monti et al., 2017).

---

[1]evaluated on the existing classical benchmark for MC Problems

### 2.2.1 THE ROW/COLUMN GRAPHS

The matrix $X$ is comprised from signals on non-euclidean domains of rows and columns. We represent those domains by undirected weighted graphs $\mathcal{G}_r$ (e.g. items) and $\mathcal{G}_c$ (e.g users) respectively, where: $\mathcal{G}_{r/c} = (V, E, W)$. $\mathcal{G}_{r/c}$ are built either directly from the ratings matrix $X$, or based on additional data about the rows/columns (if given). Their structure is encoded in Laplacian matrices which are built from the adjacency matrices $W_{r/c}$ (definitions are below). This procedure is sketched in figure 1 below.

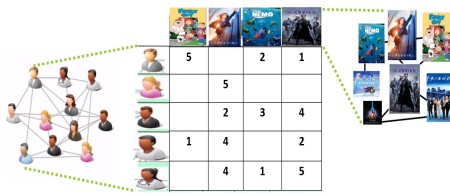

Figure 1: Example for matrix $X$ and rows/columns graphs structures

### 2.2.2 THE ADJACENCY MATRIX:

For a graph $\mathcal{G} = (V, E, W)$, the elements of its adjacency matrix $(W)_{ij} = w_{ij}$ obey: $w_{ij} = w_{ji}$, $w_{ij} = 0$ if $(i, j) \notin E$ and $w_{i,j} > 0$ if $(i, j) \in E$. The Adjacency Matrix represents the weights of the proximity between every two vertices and can be built based on the signal patterns or on external features about the rows/columns in methods like euclidean distance of normalized features, Chi square, Gaussian Kernel, K-nn clustering K-means clustering and etc.

### 2.2.3 THE GRAPH LAPLCIANS

The Laplacian matrices $L_r$ and $L_c$ are based on the adjacency matrices $W$ and are holding inside the internal Graph Structure. The most common constructions of a Laplacian matrix is an $n \times n$ matrix defined as $L = D - W$. where $D$ is degree matrix, an $n \times n$ diagonal matrix $(D)_{ii} = \sum_{j \neq i}^{n} w_{ij}$. We adopt the *Normalized Graph Laplacian* definition as $\tilde{L} = D^{-\frac{1}{2}} L D^{-\frac{1}{2}} = I - D^{-\frac{1}{2}} W D^{-\frac{1}{2}}$.

### 2.2.4 THE OPTIMIZATION PROBLEM

We use the graph laplacians in the optimization function as an additional prior regularizing the matrix completion problem. We'd like more similar items/users get more similar predictions. Mathematically, regarding the columns $x_1, \ldots, x_n$ for example as a vector-valued function defined on the vertices $V_c$, the smoothness assumption implies that $x_j \approx x_{j'}$ if $(j, j') \in E_c$. Stated differently, we want the following entity (Trace norm or Dirichlet semi-norm (Kalofolias et al., 2014)): $\sum_{i,j} w_{i,j}^c \|x_i - x_j\|_2^2 = \text{tr}\left(X L_{r/c} X^T\right)$ to be as small as possible, leading to the following optimization problem:

$$\hat{X} = \underset{x \in \mathbb{R}^{m \times n}}{\arg\min} \|P_\Omega \circ (X - M)\|_F^2 + \lambda_* \|X\|_* + \lambda_r \text{tr}\left(X^T L_r X\right) + \lambda_c \text{tr}\left(X L_c X^T\right),$$

which, if we will look at the **factorized model** will be equivalent to,

$$\hat{X} = \underset{W \in \mathbb{R}^{n \times k}, H^T \in \mathbb{R}^{k \times n}}{\arg\min} \left\|P_\Omega \circ \left(W H^T - M\right)\right\|_F^2 + \lambda_* \left\|W H^T\right\|_* + \lambda_r \text{tr}\left(W^T L_r W\right) + \lambda_c \text{tr}\left(H^T L_c H\right))$$

From this perspective, the estimation of the left and the right factors of $X$ is considered as diffusion of the input fields on the row and column graphs, respectively. This separable form allows no accommodation for the low rank constraint (which pertains to the product of the graphs).

## 2.3 DEEP NEURAL NETWORKS

In the recent years, deep neural networks and, in particular, convolutional neural networks (CNNs) (Lecun et al., 1998) based methods have been applied with great success to Image completion tasks. Such methods are based on one of the key properties of CNN architectures - the ability to extract the important local stationary patterns of Euclidean data. Image completion with untrained networks (when only the corrupted image is the input with no other training examples) can be seen as parallel to the "Matrix Completion" task. Two recent works, applying un-trained deep neural networks on corrupted images, showed state-of the art results for this task. We were inspired by those methods and our goal was to generalize them to the Non-Euclidean Domain.

### 2.3.1 DIP – DEEP IMAGE PRIOR

The method suggests to feed the network with random input Z, forward pass the random input through the network and check how close the output is to the corrupted image, while tuning the network parameters weights. This operation surprisingly reconstructs the clean image (see Ulyanov et al. (2020)).

### 2.3.2 DEEP DECODER

The Deep Decoder method showed results even better then the DIP (see Heckel & Hand (2018)). The method proposed to take a small sample of noise, and pass it through a network, while making some non-linear operations on it and up-sample, then check how far the result is from the corrupted image while fixing the network parameters. This method showed that a deep decoder network is a very concise image prior. The number of parameters it needs to completely specify that image is very small, providing a barrier for over-fitting (catching only the most important image features (natural structures) and ignore noise) and allowing the network to be amenable to theoretical analysis.

## 2.4 GEOMETRIC DEEP LEARNING OR DEEP LEARNING ON GRAPHS

In contrast to image matrices, the notion of convolution and pooling for Non-Euclidean matrices needs to be re-defined to give the Non-Euclidean stracture the special meaning that convolutional networks are based on. When those operations are redefined, we can build a "graph convolutional neural network" which is parallel to some classical neural network and find the estimate for $\boldsymbol{X}$.

### 2.4.1 CONVOLUTION FOR GRAPHS

SINGLE GRAPH CONVOLUTION: To perform a meaningful convolution operation keeping the operation translation invariant, we perform spectral graph convolution with spectral graph filters. The spectral graph theory suggest that spectral filters on matrix $\boldsymbol{Z}$ can be well approximated by smooth filters in a form of a truncated expansion in terms of Chebyshev polynomials $\boldsymbol{T}_k$ upon the rows/columns graph Laplacians $(\tilde{\Delta})$ up to some pre-defined order K (Defferrard et al., 2016). The coefficients of those polynomials $\theta_k$ are the network parameters that we learn. Using that filter approximation, we can define the Single Graph Convolution operation of a signal matrix $\boldsymbol{Z}$ with a filter $y$:

$$\boldsymbol{Z} \circledast y = \left( \sum_{k}^{K} \boldsymbol{\theta}_k \boldsymbol{T}_k \left( \tilde{\Delta} \right) \boldsymbol{Z} \right)$$

FULL MULTI GRAPH CONVOLUTION: In RMGCNN the authors propose that when a Matrix includes signals on a product of two graphs (rows/columns graph) Multi-Graph convolution is used. Full Multi Graph convolution is suitable for small matrices. When we pass signal matrix $\boldsymbol{Z}$ through a graph convolutional network layers, the output matrix after the convolution operation of matrix $\boldsymbol{Z}$ in layer $\boldsymbol{l}$ with each filter $\boldsymbol{q}$ and after adding the bios parameters $\beta q$ . ( $j$, $j'$ are the degree of the Chebyshev polynomials of the rows and columns on filters $q$ of layer $l$):

$$\widetilde{\boldsymbol{Z}}_{\boldsymbol{l}_{\boldsymbol{q}}} = \boldsymbol{Z}_{\boldsymbol{l}} \circledast y_{\boldsymbol{q}} = \left( \sum_{j,j'}^{p_l} \boldsymbol{\theta}_{jj',l\boldsymbol{q}} \boldsymbol{T}_j \left( \tilde{\Delta} \boldsymbol{r}_{\boldsymbol{l}} \right) \boldsymbol{Z}_{\boldsymbol{l}} \, \boldsymbol{T}_{j'} \left( \tilde{\Delta} \boldsymbol{c}_{\boldsymbol{l}} \right) \right) + \beta q$$

FACTORIZED MULTI GRAPH CONVOLUTION: Factorized Graph convolution is used to alleviate the computational burden for big signal matrices, using matrix factorization and then performing Single Graph convolutions on each of the factors (proposed in Bruna et al. (2014); Monti et al. (2017); Kipf & Welling (2016); Henaff et al. (2015)). It can be done in different Matrix Factorization techniques (Cabral et al., 2013; Chi et al., 2018; Zhu et al., 2018). In this model, the matrix is decomposed to it's factors with SVD: $\widehat{Z}q = \widehat{W}q\widehat{H}q^T$ . After factorizing, to get the full Multi-dimensional signal we pass each factor through the network separately and only then multiply back to get the full Multi-dimensional signal. The convolution for each factor is a **Single-Graph** convolution on each of the W and H matrices:

$$\widehat{W_q} = \sum_{j=0}^{P} \left( \boldsymbol{\theta}_j \boldsymbol{T}_j \left( \tilde{\Delta} r \right) \widehat{W_q} + \beta rq \right), \; \widehat{H_q} = \sum_{j'=0}^{P} \left( \boldsymbol{\theta}_{j'} \boldsymbol{T}_{j'} \left( \tilde{\Delta} c \right) \widehat{H_q} + \beta cq \right)$$

### 2.4.2 POOLING FOR GRAPHS

When we talk about graph pooling we talk about reducing the graph resolution grid (zoom-out operation). In our work the graph pooling is done in two steps, those steps are described below:

STEP 1: GRAPH COARSENING PROCESS (DHILLON ET AL., 2007; KARYPIS & KUMAR, 1999; SHI & MALIK, 2000)- We do bi-partition of the Graph G, to form clusters of couples. At each coarsening level, we pick an unmarked vertex i and match it with one of its unmarked neighbors j that maximizes the local normalized cut $\boldsymbol{W}_{ij} = \frac{A_{ij}}{Dii} + \frac{A_{ij}}{Djj}$ ( $A$ is graph adjacency matrix and $D$ is graph degree matrix). Then we mark the clusters as the vertices of the next level coarsened graph. The edge weight on this coarsened graph are set as the sum of the weights of the edges that connect the vertices between the clusters as described in figure 2 :

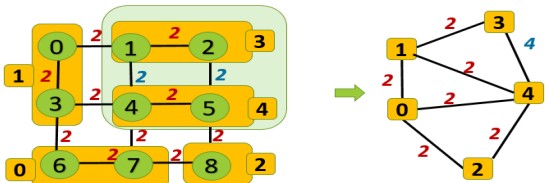

Figure 2: Graph coarsening and edge weighs assignment for the coarsened Graph

STEP 2: GRAPH POOLING / DOWN SAMPLING STRUCTURES SAVING- Defferrard et al. (2016) suggest to save the coarsening structure of the previous step in a balanced binary tree for both row/column graphs. Practically, for each row/column graph, on each layer $l$, we save in matrices $\boldsymbol{U}_{rl}, \boldsymbol{U}_{cl}$ which vertices where down sampled from which parent vertices in the previous layer. We rearrange the vertices of each adjacency matrix on each level according to this tree order, and use those matrices to constract the graph Laplacians i.e $\tilde{\Delta} r_l$ and $\tilde{\Delta} c_l$ for each network level.

### 2.5 UP-SAMPLING FOR GRAPHS

The up-sampling operation is done by multiplying the signal matrices $\hat{Z}_l$ by the parent indicators matrices $\boldsymbol{U}_{rl}$ , $\boldsymbol{U}_{cl}$ , resulting in a $\boldsymbol{Z}_{l+1}$ level matrix where the row/column parents of a specific "child" get a linear combination of its children values. We will denote this operation as:

$$\hat{Z}_{l+1} = \text{Upsample}\left(\hat{Z}_l\right) = \boldsymbol{U}_{rl}\hat{Z}_l\boldsymbol{U}_{cl}$$

***In the factorized case***, matrices $\boldsymbol{H}$ and $\boldsymbol{W}$ are up sampled separately in the same method $\widehat{W}_{l+1} = \text{Upsample}\left(\widehat{W}_l\right)$, $\widehat{H}_{l+1} = \text{Upsample}\left(\widehat{H}_l\right)$, and then multiplied to get $\boldsymbol{Z}_{l+1} = \boldsymbol{W}_{l+1}\boldsymbol{H}_{l+1}^T$

## 3  THE MATRIX DATA DEEP DECODER METHOD

In this section we present our method, the Matrix Data Deep Decoder. The core idea of our approach was to take the state of the art untrained learning model for image completion "deep decoder" and "translate" its network so it would feet for Matrix completion.

### 3.1 NETWORK PREPERATION

1. Input corrupted $m \times n$ rating matrix $\boldsymbol{M}$ and $\boldsymbol{A}_\Omega$ observation mask.
2. Input the hyper parameters: $lr$ - learning rates, $L$ - number of layers (1 - the smallest down sampled $\tilde{\Delta}\boldsymbol{r}_l$, $\tilde{\Delta}\boldsymbol{c}_l$), $Prl/Pcl$ - the degree of row/column Chebyshev polynomials on layer $l$ (i.e: the number of neighbours we'd like to consider on each layer) and finally $\boldsymbol{q}_l = q_1, \ldots, q_L$ - number of neurons on each layer $l$. Each nueron learns $Prl/Pcl$ coefficients of the Chebyshev polynoomials of the row/column Laplacians.
3. Build Initial row/cols graph adjecency matrices $(\boldsymbol{Ar}_L, \boldsymbol{Ac}_L)$, based on: $\boldsymbol{M}$, row/columns properties and selected distance function (we used the threshold clustering: for every couple of rows/columns, saving the Euclidean distance between their attributes only if it is smaller then threshold $R$, otherwise 0. Alternatively, for specific data combinations of rating patters and user/item attributes can be taken as featurs.)
4. Build Initial row/cols normalized graph Laplacians $\left( \tilde{\Delta}\boldsymbol{r}_l, \tilde{\Delta}\boldsymbol{c}_l \right)$ based on $(\boldsymbol{Ar}_L, \boldsymbol{Ac}_L)$
5. For each $l = L - 1, \ldots 1$ **, for each rows/col** graph adjacency matrices do:

    5.1. Build the rows/cols coarsed edge weights matrix $(\boldsymbol{W}_l)_{ij} = \frac{(\boldsymbol{A}_l)_{ij}}{(\boldsymbol{D}_l)_{ii}} + \frac{(\boldsymbol{A}_l)_{ij}}{(\boldsymbol{D}_l)_{jj}}$
    $\boldsymbol{A}_l$ is the graph adjacency matrix and $\boldsymbol{D}_l$ is the graph degree matrix.
    5.2. Cluster couple of every two closest vertices to $\boldsymbol{W}_l$ distance and enumerate the clusters as the new graph vertices in pooling matrices $\boldsymbol{Ur}_l$, $\boldsymbol{Uc}_l$
    5.3. Build reduced adjacency matrices $\boldsymbol{Ar}_l$, $\boldsymbol{Ac}_l$ when the matrix instances (edges weights) are the sum if the distances between the clusters members in matrix $\boldsymbol{W}_l$ as described in figure 2 and rearrange in the order of $\boldsymbol{Ur}_l$, $\boldsymbol{Uc}_l$.
    5.4. Build the row/cols normalized graph Laplacians $\left( \tilde{\Delta}\boldsymbol{r}_l, \tilde{\Delta}\boldsymbol{c}_l \right)$ based on $\boldsymbol{Ar}_l$, $\boldsymbol{Ac}_l$

6. Take a matrix $\widehat{\boldsymbol{Z}}_1$ in the size of $\left( \tilde{\Delta}\boldsymbol{r}_1 \times \tilde{\Delta}\boldsymbol{c}_1 \right)$ and input (infuse) a random noise in it.

### 3.2 NETWORK LEARNING PROCESS ALGORITHMS

---

**Algorithm 1 (MDDD)**

---

**For** $t = 0 : T$ do:

*Network Learning process Forward pass:*

**1.** for $\ell = 1 : L - 1$ (last layer with the smallest $\tilde{\Delta}\mathbf{r}_\ell, \tilde{\Delta}\mathbf{c}_\ell$) do:

   1.1 *perform* for matrix $\widehat{\boldsymbol{Z}}_\ell$ Full Multi-Graph convolution:

$$\widehat{\boldsymbol{Z}}_\ell \leftarrow \sum_{k=0}^{q_\ell} \sum_{j,j'=0}^{p_r,p_c} \boldsymbol{\theta}_{j,j',k} \boldsymbol{T}_j \left( \tilde{\Delta}\mathbf{r}_\ell \right) \widehat{\boldsymbol{Z}}_\ell \boldsymbol{T}_{j'} \left( \tilde{\Delta}\mathbf{c}_\ell \right) + \beta_k$$

   1.2 Apply a ReLu non-linearity: $\widehat{\boldsymbol{Z}}_\ell \leftarrow \boldsymbol{\xi} \left( \widehat{\boldsymbol{Z}}_\ell \right)$
   1.3 $\widehat{\boldsymbol{Z}}_{\ell+1} = Upsample \left( \widehat{\boldsymbol{Z}}_\ell \right)$ (c.f sec. 2.5)

**2. End for**

**3.** Normalize: $\widehat{\boldsymbol{Z}}_L = \mathbf{Sigmoid} \left( \widehat{\boldsymbol{Z}}_L \right)$

*Network Backward Pass:*

**4.** Update the parameters $\theta$ and $\beta$ for all layers with gradient descent, minimize the loss between $\widehat{\boldsymbol{Z}}_L$ and the observed $\boldsymbol{M}$:

$$\widehat{\boldsymbol{X}}_t = \underset{\boldsymbol{X} \in \mathbb{R}^{m \times n}}{\mathrm{argmin}} \left\| \boldsymbol{A}_\Omega \circ \left( \widehat{\boldsymbol{Z}}_L - \boldsymbol{M} \right) \right\|_F^2 + R \left( \widehat{\boldsymbol{Z}}_L \right)$$

$$R \left( \widehat{\boldsymbol{Z}}_L \right) = \lambda_* \left\| \widehat{\boldsymbol{Z}}_L \right\|_* + \lambda_r \mathrm{tr} \left( \widehat{\boldsymbol{Z}}_L^T \tilde{\Delta}\mathbf{r}_L \widehat{\boldsymbol{Z}}_L \right) + \lambda_c \mathrm{tr} \left( \widehat{\boldsymbol{Z}}_L \tilde{\Delta}\mathbf{c}_L \widehat{\boldsymbol{Z}}_L^T \right)$$

**5.** $RMSE_t = \sqrt{\frac{\left\| \boldsymbol{T}_\Omega \circ \left( \widehat{\boldsymbol{X}}_t - \boldsymbol{M} \right) \right\|_F^2}{\Sigma_{i,j} \, \boldsymbol{T}_{i,j}}}$ ($\boldsymbol{T}_\Omega$–test observation mask)

**End For.** Return matrix $\boldsymbol{X}^* = \widehat{\boldsymbol{X}}_i$ where $RMSE_i$ is the smallest

---

**Algorithm 2 (sMDDD)**

---

**For** $t = 0 : T$ do:

*Network Learning process Forward pass:*

**1.** for $\ell = 1 : L - 1$ do:

   1.1 Factorize $\widehat{\boldsymbol{Z}}_\ell$ with SVD $\boldsymbol{W}_\ell$ (rows) and $\boldsymbol{H}_\ell$ (cols.)

   1.2 *perform* Factorized Multi-Graph convolution:

$$\widehat{\boldsymbol{W}}_\ell \leftarrow \sum_{k=0}^{q_\ell} \sum_{j=0}^{p_{r\ell}} \boldsymbol{\theta}_{j,k} \boldsymbol{T}_j \left( \tilde{\Delta}\mathbf{r}_\ell \right) \widehat{\boldsymbol{W}}_\ell + \beta_k, \widehat{\boldsymbol{H}}_\ell \leftarrow \sum_{k=0}^{q_\ell} \sum_{j=0}^{p_{c\ell}} \boldsymbol{\theta}_{j,k} \boldsymbol{T}_j \left( \tilde{\Delta}\mathbf{c}_\ell \right) \widehat{\boldsymbol{H}}_\ell + \beta_k$$

   1.3 Apply a ReLu non-linearity: $\widehat{\boldsymbol{W}}_\ell \leftarrow \boldsymbol{\xi} \left( \widehat{\boldsymbol{W}}_\ell \right), \widehat{\boldsymbol{H}}_\ell \leftarrow \boldsymbol{\xi} \left( \widehat{\boldsymbol{H}}_\ell \right)$
   1.4 $\widehat{\boldsymbol{W}}_{\ell+1} = Upsample \left( \widehat{\boldsymbol{W}}_\ell \right), \widehat{\boldsymbol{W}}_{\ell+1} = Upsample \left( \widehat{\boldsymbol{H}}_\ell \right)$

**2. End for**

**3.** Pass through Fully Connected Layer ($\boldsymbol{\theta}_{wfc}, \boldsymbol{\theta}_{hfc}$ have $\widehat{\boldsymbol{W}}_L, \widehat{\boldsymbol{H}}_L$ shapes):
$\widehat{\boldsymbol{W}}_L \leftarrow \boldsymbol{\theta}_{wfc} \widehat{\boldsymbol{W}}_L + \beta_{wfc}, \widehat{\boldsymbol{H}}_L \leftarrow \boldsymbol{\theta}_{hfc} \widehat{\boldsymbol{H}}_L + \beta_{hfc}$

**4.** Normalize: $\widehat{\boldsymbol{Z}}_L = \mathbf{Sigmoid} \left( \widehat{\boldsymbol{W}}_L \widehat{\boldsymbol{H}}_L^T \right)$

*Network Backward Pass:*

**5.** Update parameters ($\theta, \beta$) for all layers with gradient descent to minimize loss:

$$\widehat{\boldsymbol{X}}_t = \underset{\boldsymbol{X} \in \mathbb{R}^{m \times n}}{\mathrm{argmin}} \left\| \boldsymbol{A}_\Omega \circ \left( \widehat{\boldsymbol{Z}}_L - \boldsymbol{M} \right) \right\|_F^2 + R \left( \widehat{\boldsymbol{Z}}_L \right)$$

$$R \left( \widehat{\boldsymbol{Z}}_L \right) = \frac{\lambda_*}{2} \left\| \widehat{\boldsymbol{Z}}_L \right\|_* + \lambda_r \mathrm{tr} \left( \widehat{\boldsymbol{W}}_L^T \tilde{\Delta}\mathbf{r}_L \widehat{\boldsymbol{W}}_L \right) + \lambda_c \mathrm{tr} \left( \widehat{\boldsymbol{H}}_L^T \tilde{\Delta}\mathbf{c}_L \widehat{\boldsymbol{H}}_L \right)$$

**6.** $RMSE_t = \sqrt{\frac{\left\| \boldsymbol{T}_\Omega \circ \left( \widehat{\boldsymbol{X}}_t - \boldsymbol{M} \right) \right\|_F^2}{\Sigma_{i,j} \, \boldsymbol{T}_{i,j}}}$ ($\boldsymbol{T}_\Omega$–test observation mask)

**End For.** Return matrix $\boldsymbol{X}^* = \widehat{\boldsymbol{X}}_i$ where $RMSE_i$ is the smallest

---

### 3.3 STOPPING CRITERIA

In this work we run the algorithm for each set for T=10000 iterations as in RMGCNN. We use the Iteration weights on which we've obtained the best RMSE for the test set. Thus, the number of iterations is another hyper parameter that should be tuned for best performance.

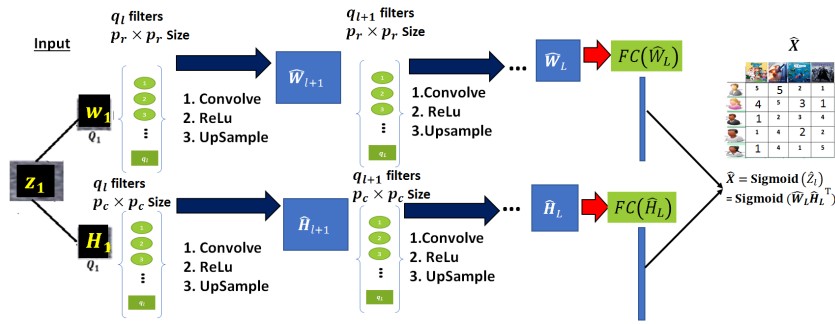

Figure 3: Network Structure Illustration – MDDD algorithm flow scatch

## 4   EXPERIMENTAL RESULTS

Up to date, the state of art results for Matrix Completion where obtained by the RMGCNN method, hence, in the experimental results we compare our results to are the results of the RMGCNN. The information about the other methods was taken from the RMGCNN work and can be found in the work of [30].  Our tested datasets started with a small simple synthetic dataset – Synthetic Netflix. Then we have continued by evaluating our method on real banchmark datasets: MovieLens 100k, Flixster, Douban and YahooMusic following Monti et al. (2017). The tested datasets statistics described in table 1 below:

Table 1: Datasets Statistics. "Graphs"=rows/columns availability for graphs construction

| Dataset | #Users | #Items | Graphs | #Ratings | Density | Rating levels |
|---|---|---|---|---|---|---|
| ML-100K | 943 | 1682 | Users/Items | 100,000 | 0.0630 | $1, 2, \cdots, 5$ |
| Douban | 3000 | 3000 | Users | 136,891 | 0.0152 | $1, 2, \cdots, 5$ |
| Flixster | 3000 | 3000 | Users/Items | 26,173 | 0.0029 | $0.5, 1, \cdots, 5$ |
| YahooMusic | 3000 | 3000 | Items | 5,335 | 0.0006 | $1, 2, \cdots, 100$ |

### 4.1   BENCHMARK RESULTS ON THE TESTED DATASETS

In this section we describe the data preparation parameters of the MDDD method for all tested datasets and then present in a summerizing tables the Results compared to other known methods.

THE NETFLIX CHALLANGE-    On the Synthetic Netflix dataset we tested both, the Non-Factorised and the factosised algorithms.  In MDDD we used row degree 1,column degree 5.  The graphs where constructed using 10 nearest neighbours.  Our Non-Fuctorised MDDD model achieved the best accuracy, followed by our Factorised MDDD model as described in table3 (a) below .

THE MOVIELENS, FLIXTER, DOUBAN AND YAHOOMUSIC CHALLENGE-    For the real datasets, Movielens_100k, Flixter, Douban and YahooMusic the network was prepared as following: For all datasets only the MDDD Factorised model was used, due to the big datasets size.  In all settings learning rate was set to 0.01. and fully connected layer was used to translate the rows and columns embedding to the Ratings space.  For all datasets the training and tests sets where taken exactly as in Monti et al. (2017). For the Movielens_100k Dataset (the most familiar benchmark dataset), the rows/column graphs where built using the threshold method described in section 3.2.2. The network had 2-layers with 32 filters on each layer and filter size of 3x3.  For all other datasets, we have used the same graphs that were used in the RMGCNN work [30].  As for the YahooMusic/Douban datasets only columns or rows features were available respectively (tracks fatures or users features), to build the missing Graphs for those databasets we used the training set of the corrupted matrix with 10-nn neighbors.  For example for Douban, we have used 7.5% of the ratings that we've marked as the training set.  In the Douban dataset, we used a 2-layer network, with 64 filters on each layer, of size 2x2.  In the Flixter dataset, we used a deeper network of 4 layers (64,64,32,32) filters on each with filter sizes of (1x1,40x40,1x1,1x1) on each layer. Table 2 summarizes the performance of different methods compared to MDDD.

Table 2: Performance (RMS error) comperisson

(a) Comparison of different matrix completion methods on Synthetic Netflix in terms of number of parameters (optimization variables) and computational complexity order (operations per iteration). Rightmost column shows the RMS error on Synthetic dataset.

| METHOD | PARAMETERS | COMPLEXITY | RMSE |
|---|---|---|---|
| GMC | $mn$ | $mn$ | 0.3693 |
| GRALS | $m + n$ | $m + n$ | 0.0114 |
| RGCNN | 1 | $mn$ | 0.0053 |
| sRGCNN | 1 | $m + n$ | 0.0106 |
| **MDDD (Factorized)** | **1** | **$m + n$** | **0.0017** |
| **MDDD (Multi-Graph)** | **1** | **$m + n$** | **0.0004** |

(b) Performance (RMSE) of different matrix completion methods on the MovieLens 100k dataset.We got the best results with a 2-layer network when each layer had 64 parameters; the 1st layer had 1x1 filters and the 2nd 40x40.

| METHOD | RMSE |
|---|---|
| GLOBAL MEAN | 1.154 |
| USER MEAN | 1.063 |
| MOVIE MEAN | 1.033 |
| MC (CANDES & RECHT, 2012) | 0.973 |
| IMC (JAIN & DHILLON, 2013; XU ET AL., 2013) | 1.653 |
| GMC (KALOFOLIAS ET AL., 2014) | 0.996 |
| GRALS (RAO ET AL., 2015) | 0.945 |
| sRGCNN | 0.929 |
| **MDDD** | **0.922** |

(c) Performance (RMSE) on Douban, Flixster/Flixter-U (when only users Graph Exists) and YahooMusic with only ratings Graph. Other baselines are taken from 32

| METHOD | DOUBAN | FLIXSTER/Flixter-U | YAHOOMUSIC |
|---|---|---|---|
| MC | 0.845 | 1.533/ 1.534 | 52.0 |
| GRALS | 0.833 | 1.313/ 1.243 | 38.0 |
| GC-MC | 0.734 | 0.917/ 0.941 | 20.5 |
| sRMGCNN | 0.801 | 1.179/ 0.926 | 22.4 |
| **MDDD** | **0.733** | **0.893** | **20.2** |

## 4.2 RESULTS DISCUSSION

For the Synthetic Netflix dataset our Non-Fuctorised MDDD model achieves the best accuracy, followed by our Factorised MDDD model (table 2 (a)) . For the real datasets (Movielens, Flixter, Douban and YahooMusic) , MDDD (the Factorized Model) outperforms the competitors (Monti et al., 2017; Rao et al., 2015; Yao & Li, 2018) in all the experiments (table 2 (b),(c). Our algorithm also gets the result in much less running time.For example On the Movielens dataset our algorithm converges after 1800 Iterations, compared to 25,000 in RMGCNN algorithm ( 5 minutes compared to 30 minutes - (table 2 (b))). The algorithm has shown an improvement in state of the art result in 7% , and can be farther improved (appendix A and B) but most importantly, we got the results very quickly. We consider the reason for that the model simplicity and the small amount of parameters that make it easier for the algorithm to first re-construct the natural graph structures, then the noise.

## 5 CONCLUSIONS

In this work we addressed the problem of Matrix Completion on Non-Euclidean domains, where sparse signal, which lies on a grid of two non- Euclidean domains (Graphs or manifolds), should be completed. We introduced a new method for the Matrix Completion Problem solution: The Matrix Data Deep Decoder - a simple, intuitive under-parametrized yet powerful method for Matrix Completion inspired by the Deep Decoder method for Image Completion. As far as we know this is the 1st method for non-Euclidean matrix data completion that is end to end based on fully convolutional network. Despite it's simplicity the method shows state of art results on current known benchmarks in both predictions error ( 7% improvement) and runnig time in ( 6 times faster). Because of the method simplicity, it can be applicable in variety of fields and real life problems like recommendations systems (the Netflix Problem), pattern recognition, community detection Biological consequences on gene data or DNA structure, Chemical reactions, Physical applications , Events prediction, traffic detection, stocks prediction and many more. It can also be expended to higher dimensional spaces like tensors instead of matrices and new research directions (appendix A and B).

Our method can suggest that when we are looking at the problem of Matrix Completion from the geometric point of view (as a sparse signal that lies on underlying rows/columns or their product graphs structures), convolutional neural networks can use a very strong prior for that problem solution. For future research and applications it means a continuous improvement in the field of Non-Euclidean learning networks, in parallel to the improvement in the field of the classical learning networks.

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

## A APPENDIX A

FUTURE WORK In this work we have presented the results of the described method that was the basis for us in order to get new state of the art results. But, we do believe that the results can be farther improved by the following future research : tuning he hyper parameters, using novel methods for Graph Convolution on the convolutional layers for example different graph coarsening (Liu et al., 2019), using methods for learning the best Metric for the initial Adjacency Matrices and the Laplacians in the same or separate network, using Multi-Graph convolution for the smaller network layers and factorized convolution on the bigger levels, using other different methods for matrix factorization and using Product Graph Laplacians instead of the row/columns (Imrich & Klavzar, 2000; Ortiz-Jiménez et al., 2018).

.

In future work we also believe that there can be an expansion of the results to a higher dimentional space (from matrix to tensor completion). One of our side experiments was testing the method on a real live mobile adds platform that shared Terabytes of information with us, and included a tensor of User(gamer), Content(game) and Context(page appearance and details in which the game advertised), and got prediction results for the content downloading much better and faster then RMGCNN and better then every company algorithm.

The data included: The shared data included the following files: a. Apps - 183,199 Applications and Host Applications. They included different properties of the application like genre, developer details, price, OS and etc.

b. Hosts – like Apps – the same 183,199 Applications and Host Applications. They included different properties of the application like genre, developer details, price, OS and etc.

c. Users - 12,160,088 Users that included users' properties like their origin, country, locale, interests and other user specific properties.

d. Events - 632,849,289 Events that included the user, application, context (application host), and the specific event that occurred by the user (impression, click, install and etc.).

We have predicted the Clicks of users on the apps – but it could work with any other kind of event.

The results were as following:

Table 3: Results comparison on a real live Adds Dataset.

| METHOD | RMSE | Time to best result |
|---|---|---|
| sRMGCNN | 0.0172 | ~ 30 min |
| **MDDD** | **0.0160** | **~5 min** |

In addition, Dynamic inference (in which the matrix itself describes a time process and the geometry of the row/column spaces has some non-trivial dynamics) can be a great extension to the algorithm applications but requires a separate future research in different settings and benchmarks .

Finally, in future work we intend and strongly recommend to other scientists to explore if new Learning methods that are state of art in the field of single image completion problem (in the 2-D Euclidean domain) may yield new state of art results also for the solution of the Parallel Non-Euclidean Matrix Completion problem in the way we have translated the Deep Decoder network in this article, for Non-Euclidean Domains.

## B    APPENDIX B

SIDE RESEARCH - SMALL MATRICES COMPLETION IN PRODUCT SPACE    One of the experiments that we did as part of our work was to test the idea of matrix completion when we input not only the ratings matrix, but the whole product space. To test this, we took the Movielens 100k database and created random cuts from the matrix $X$ of 100x100 size. We took 30% of the data as the observed data and 7.5% of the observed data as the training data. Our goal was to complete the whole 100x100 ratings matrix (see example run in figure 5). First we ran the RMGCNN algorithm as described in Monti et al. (2017) . Then we ran the same algorithm on the product space instead, meaning, that our input was the full product space matrix (as described in figure 4). In all experiments we constructed the Laplacian matrices using only the features of users/movies and a partial rating column. When we compared the RMSE in all cuts,the result was a higher RMSE of 20-40 % in case we inputted the whole product space (see table 4 for all cuts). In addition, not only the ratings where completed but also all other features (like age, gender etc.). The drawback of this method was that the learning took a long time and the time grows exponentially with every user/item couple. The advantage of this method is that it can be applicable for real life instances when small sparse matrices with important data should be completed (like small businesses or detective purpose etc.). It also worthwhile to test different estimators for the product space. Future research in this direction might yield astonishing results.

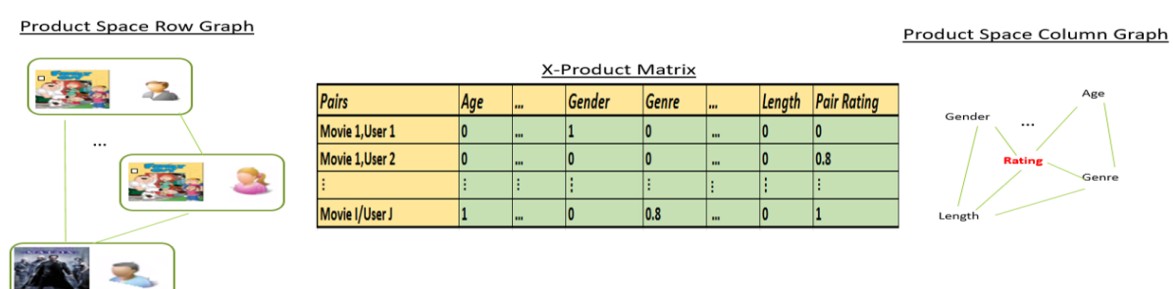

Figure 4: The full product space matrix input illustration

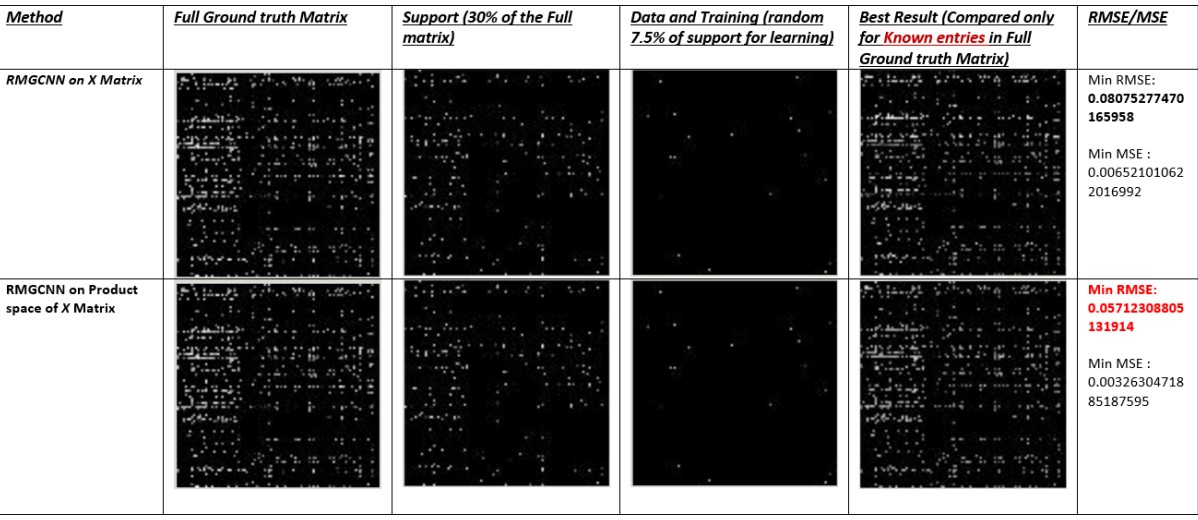

Figure 5: Example run illustration - The cut of users 600:700 and movies 600:700

Table 4: All 10 cuts RMSE comparison

| | Run1 | Run2 | Run3 | Run4 | Run5 | Run6 | Run7 | Run8 | Run9 | Run10 | RMSE Average | RMSE Variance |
|---|---|---|---|---|---|---|---|---|---|---|---|---|
| Matrix X input | 0.1333 | 0.10558 | 0.13624 | 0.09713 | 0.09886 | 0.07926 | 0.08075 | 0.06518 | 0.05351 | 0.12977 | 0.097958577 | 0.000831246 |
| Product space Input | 0.10715 | 0.08886 | 0.10705 | 0.07225 | 0.07756 | 0.06015 | 0.05712 | 0.04864 | 0.04448 | 0.10247 | 0.076571576 | 0.000573632 |
| Average Improvement | 24.41% | 18.82% | 27.27% | 34.44% | 27.47% | 31.78% | 41.37% | 34.00% | 20.31% | 26.65% | 28.65% | |

