# OpenReview forum: "Matrix Data Deep Decoder - Geometric Learning for Structured Data Completion"
_ICLR.cc/2021/Conference — Reject_

### Official Review · AnonReviewer4 · 2020-10-22
**The main contribution of this paper is to adopt matrix data deep decoder for matrix completion task.**

**Rating:** 3
**Confidence:** 4

**Review:**

In this work,  the authors addressed the problem of Matrix Completion on Non-Euclidean domains.  They mainly adopt the method Matrix Data Deep Decoder, inspired by the Deep Decoder method for Image Completion. Here are my main concerns:

1. The technical contributions of this paper is limited since it mainly adopts existing methods, like Matrix Completion + Deep Decoder + multi-graph convolution. It seems for me this paper is more like an engineering paper consist of lots of stuff. It is better to release the source code to understand this kind of composition stuffs.

2. The motivation of this work is also not very sound. The authors spend lots of content on the related work. However, the main idea is only presented in a presuppose-code style in Page 6, which is not very clear.

3. In the experimental parts. Why use different methods for different dataset in Table 2(a), (b), (c)?  Is it possible to conduct all of the baselines for all datasets? Also,  the matrix completion has been studied for many years with many solid theory results,  the baselines in the experiments are not SOAT.

Typos:
In page 4: stracture -------> structure
In page 7:  can be found in the work of [30] -----> the cite form is not correct.

In summary, the paper show that a GCN network is a good prior for the Matrix completion problem. However,  the technical contributions of this paper is limited and the overall presentation can be further improved.

---

### Official Review · AnonReviewer3 · 2020-10-28
**Nontitle's**

**Rating:** 3
**Confidence:** 4

**Review:**

It could be accepted with major changes in the structure of the paper and rewriting some parts. But I would prefer to vote for rejecting and ask them for upgrading the writing overall.

Pros:
Very promising method and concept.
Experimental results look great comparing to the two state-of-the-art methods (Referring to table 2 – performance comparison)

Cons:
1.	The grammar and style of writing are not as high as expected.
Several typos and lowercase/uppercase misuse throughout the whole paper. Numerous grammatical errors.
Use of several run-on sentences. For example:
In abstract:
Use of the adjectives “surprising” and “spectacular” for the state-of-the-art methods is uncustomary language for a technical literature.
The sentence starting with “In sharp contrast to previous …” is a run-on sentence. It can be broken to at least two sentences.
Then the sentence containing “getting that way state of the art …” is vague. Do you mean “in that way we get?”
The sentence starting with “In addition to the conceptual simplicity of our method …” is a run-on sentence. Should be broken to at least two sentences.

In section 1 Introduction:
“In this work, we present a simplified …” sentence is too long.
After “a classical end to end graph convolutional neural network and is/was inspired…”, is/was is missing before inspired.
The sentence starting with “When, the geometry of the column/row …” is grammatically incorrect. It could be connected to the previous sentence with a phrase in which, instead of using “when”.

2.	The structure of the paper is not completely standard.
•	Contribution i and ii are technically one contribution and should be merged.
•	In section 2.3 “two recent works” are mentioned with no immediate reference to the papers.
•	Future work could be a section after a conclusion called future work, not as an appendix. The appendix is usually for detailed descriptions of methods.
•	Too many small subsections.
o	Too many small subsections in section 2. Preliminaries.
o	In section 3.1, network preparation starts with bullet points right away. An explaining sentence or two is needed before starting the bullet point. The same issue at 3.2, only two figures are shown with no extra explanation.
o	3.3 is not a necessary subsection. Should be merged with other explanations of this section.

---

### Official Review · AnonReviewer2 · 2020-10-28
**Official Blind Review #2**

**Rating:** 4
**Confidence:** 4

**Review:**

This paper aims to tackle the matrix completion problem by drawing connection from prior work in image completion domain. It seems to be a combination of prior work: Multi-graph convolution combined with Dirichlet energy on row and column graph laplacian where the input rating matrix is corrupted with noise. The writing and presentation is significantly below par Iclr acceptance in the current form. Also, considering some of the work mentioned below, SOTA results is an overclaim.

a) Clarity Issues-
Page 6 is incomprehensible in current form where the main algorithm is described. There are too many font changes in the results and main algorithm section. There is a grammatical / spelling error or typo almost every 5 lines throughout the paper. Few examples below:

-results for Matrix Completion *where* obtained
-Figure 3: Network Structure Illustration – MDDD algorithm flow *scatch*
-then present in a *summerizing* tables the Results compared to other known methods.
-the Non-Factorised and the *factosised* algorithms.
-*translate" its network
-so it would *feet* for Matrix completion

b) State of the art on Matrix completion:
While this work claims state of the art results, it is missing some recent work (Iclr 20) that have achieved better state of the art results than reported in this paper.

1) Inductive Matrix completion with gnn iclr 20 https://openreview.net/forum?id=ByxxgCEYDS
2) Deep Matrix Factorization with spectral regularizers (https://arxiv.org/abs/1911.07255)

c) Related work :
This section is completely missing. While authors use 3 and half pages to describe the background needed,  it is not clear to me how this work uses multi graph convolution used in prior work or how does the authors make use of Dirichlet energy that is defined in preliminary section if it is differently used here than prior work.

---

### Official Review · AnonReviewer1 · 2020-10-28
**Contribution Limited; Lacking Theoretical Understanding**

**Rating:** 3
**Confidence:** 5

**Review:**

Topic: Matrix Completion for Recommender Systems


main contribution

using the idea of `deep prior for matrix completion.


Strength

-  performance: It seems that the method outperforms a number of baselines on different recommender system tasks. This suggests the usefulness of the proposed method.



Weakness

- Contribution Limited: The problem is formulated as graph Laplacian regularized matrix completion problem. This has been a long-existing technique. The contribution of this work lies in modeling the complete matrix within the range of a deep generative model and learns the deep generative model on the fly. GNN based link prediction is not a new technique. The work puts GNN based link prediction together with low-rank regularization, which is perhaps the reason why some practical benefits were observed, but the contribution is limited.

In fact, it is very hard to see how much new contribution is there on top of the main reference

Federico Monti, Michael M. Bronstein, and Xavier Bresson. Geometric matrix completion with
recurrent multi-graph neural networks. CoRR, abs/1704.06803, 2017. URL http://arxiv.
org/abs/1704.06803.

There may have been some regularization and network modifications, but the new contributions seem to be marginal.

Is it possible to clarify what are the new contributions on top of the framework from the above reference?


- Clarity and Soundness:
 -- Most of the effort was invested onto constructing the graph CNN that is claimed tailored for non-Euclidean matrices. But this part looks a bit theoretically (or even intuitively) ungrounded --- particularly, it is not easy to follow the rationale behind sec. 2.4. It is also unclear how is this construction suitable for handling non-Euclidean cases. This part may need some more explanations.

-- the writing of the work is a bit convoluted or at least misleading. It tries to connect with deep prior based approaches, but the actual method is not really related. In a nutshell, the approach is a regression approach that maps embeddings of the row and column entities of the matrix to be completed to the ratings. The embeddings are constructed from graph Laplacians, and the mapping (regression model) is a particularly constructed neural network. It is suggested that these points to be clearly articulated.


- Technical Depth: The regression formulation with the regularization and constraints are not outright unreasonable, but it seems that there is no further justification of the approach beyond intuition. The reason why the regression model/network should be constructed this way other than other ways is unclear and unjustified, even on an intuition level. The technical depth of this work may have large room to improve.

- Simulation validation: It is unclear if the proposed algorithm converges or not, even on simple simulated data. It is suggested to show the algorithms’ numerical behaviors so that the readers have some sense about its convergence characterizations. The current version does not have such information.

- Questionable Mathematical Representation: The last equation in page 3 looks a bit strange: if X is approximated by WH’, then it is automatically rank limited (by the number of columns of W,H). Adding a nuclear norm on WH’ looks unnecessarily complicated.


- writing: The writing seems to have been done in a hastily manner, with many typos (see “Minor”).





Minor:

page 2: ``non-convex (but still very well-behaved)’’: what does it mean?
page 2: two equations, “min” should be “arg min”?
page 4  sec 2.4 “stracture”
sec 2.3.2 “better then”
most of the tables and figures seem to have very low resolution
sec 3.1 is very hard to read through

---

### Decision · Program_Chairs · 2021-01-07
**Final Decision**

**Decision:**

Reject

**Comment:**

The focus of this paper is to analyze an end to end network to reconstruct matrices originating from non-Euclidean data which are corrupted. The authors present an untrained network for this task. In the review period the reviewers raised a variety of concerns including concerns about novelty of the paper with respect to existing work, technical depth and clarity. The authors did not respond to these concerns. Therefore, I recommend rejection.